# Comparison of Wind Lidar Data and Numerical Simulations of the Low-Level Jet at a Grassland Site

**Astrid Ziemann** [1,*] **, André Galvez Arboleda** [2] **and Astrid Lampert** [2]

1    Institute for Hydrology and Meteorology, TU Dresden, Pienner Str. 23, 01737 Tharandt, Germany
2    Institute of Flight Guidance, TU Braunschweig, Hermann-Blenk-Str. 27, 38108 Braunschweig, Germany; andregalvez10@hotmail.com (A.G.A.); astrid.lampert@tu-braunschweig.de (A.L.)
*    Correspondence: astrid.ziemann@tu-dresden.de

**Abstract:** For the increasing importance of the wind energy branch, exact wind climatologies at the operation altitudes are essential. As wind turbines of increasing hub height are erected, the rotors are located at an altitude interval influenced by the phenomenon of low-level jet (LLJ). The main objective of the study is to assess if and how numerical simulations can represent the development especially of nocturnal LLJs in comparison to measurements. In this article, the microscale numerical model HIRVAC2D is used for a range of parameters. The simulated results for properties of the LLJ are compared to lidar data at an altitude range of 40 m to 500 m at the study site Braunschweig in the North German Plain, a grassland location that may be representative for a large area. Similarities and differences of the occurrence, height and maximum wind speed of the nocturnal LLJ are discussed using two different criteria to define a LLJ. The analysis of the lidar data set for the grassland site revealed for the first time increasing height of the LLJ with increasing wind speed during the summer months June to August 2013. The comparison of measurements and simulation data shows that boundary (and inital) conditions have to be adapted in model simulations to provide realistic LLJ properties. It was found that land use and vegetation parameters are important for practical LLJ prognosis, both for wind climatologies and nowcasting.

**Keywords:** low-level jet; wind lidar; micro-scale model simulation; HIRVAC2D

## 1. Introduction

In Germany, the contribution of renewable energies to the overall power consumption is increasing continuously to 17% in 2019 [1]. A large part of the renewable energy is produced by wind parks. Future perspective of wind energy is embedded in the program of the European Union (The European Green Deal) for decarbonizing the energy sector to reach the climate objectives in 2030 and 2050. In this context renewable energy sources, especially the extension of the offshore wind sector will play an essential role [2]. On global scale there are different analyses showing that wind energy potential is able to cover more than one third of the global power in 2050 [3]. Above land, wind turbines with a total capacity of 50.6 GW were installed in 2017 [4]. Onshore wind turbines have a typical hub height from 80 m to 140 m and a rotor diameter of 80 m–118 m [4]. There are large international projects to improve the knowledge of the resource wind speed, e.g., the New European Wind Atlas (NEWA), providing high resolution data of wind speed in Europe based on model downscaling and observations (e.g., [5,6]).

In this altitude range, a specific wind phenomenon occurs regularly, the low-level jet (LLJ). The LLJ is a wind maximum in the vertical profile of horizontal wind speed. Criteria to detect a LLJ vary according to the application [7], there is no unique definition [8]. The main mechanism of low-tropospheric LLJ development considered in this study is the decoupling of surface friction

from the atmospheric boundary layer (ABL) by the formation of a temperature inversion [9]. This is typically the case during night, in particular on days with few cloud cover and strong radiative cooling. The LLJ typically develops around sunset, reaches maximum intensity in the early morning hours, and decays with the onset of vertical mixing during daytime. It is characterized by a pronounced, supergeostrophic wind speed maximum, frequently at height levels less than 500 m above ground [10]. Climatologies of LLJ indicate a frequency of occurrence of about 10 to 50% of all nights, depending on location and on the applied criteria [7,11].

Marke et al. [12] confirmed the importance of LLJs for wind power applications: Numerous LLJs (16% of all LLJ) have the wind speed maximum in the range of wind-turbine rotor heights below 200 m. A precise forecast and uncertainty quantification of the expected wind speed are crucial to numerous wind power applications [13], e.g., to estimate the power output for the identification of promising sites for wind power generation. Another application is real-time wind power production forecasting. This is the base for financial calculations of operators, and very important for grid planning [4]. Therefore, suitable predictions of the LLJ strength and altitude play an important role [14]. The increased wind speed during LLJ events may contribute to increased power output [15–17]. On the other hand, wind shear, both concerning wind speed and wind direction, imposes additional load on the rotor blades and shifts wind turbine vibrations to higher amplitudes [18]. The increased wind shear associated with LLJs induces turbulence, which in turn counteracts wake effects [19].

On this scale between the meso- and the microscale it is particularly challenging to understand and numerically describe flow processes [20]. Meanwhile, several possibilities are applied to simulate LLJs: analytical models [10,21] or conceptual views [22], single-column models [23], mesoscale models like the Weather Research and Forecasting WRF [13] or Large-Eddy Simulation (LES) models [12].

Several studies investigated the numerical simulation of LLJs by mesoscale models at locations of the Great Plains. Storm et al. [24] and Steeneveld et al. [25] found that WRF model runs underestimated LLJ strength and overestimated LLJ height. Mirocha et al. [13] observed significant differences between lidar measurements and WRF simulations. Lidar data have been used for data assimilation to improve the mesoscale forecast of wind speed [26]. The model resolution especially in vertical direction is one factor influencing the numerical simulation of LLJs due to the existing large gradients in wind speed and direction. A further and perhaps the most important aspect is the kind of ABL modelling including the parameterization of the near-surface exchange between soil, vegetation and atmosphere that is important for the realistic representation of the LLJ [13].

Against this background, the applicability of a prognostic microscale ABL model for LLJ climatologies and wind power forecasting was tested in this study. The used model is characterized by a high spatial resolution of the vegetation. In this way, the influence of land use on the unsteady wind profile can be described via the realistic exchange of momentum and heat. The aim of this article is to compare numerical simulations of the LLJ with wind data of a wind lidar operated at the site Braunschweig in the height range of 40 m to 500 m and indicate potential for model development.

This article is built in the following way. Section 2 describes the simulated and the measured data sets analysed in this study, together with the criteria of LLJ identification and the model setup. The results of the analysed wind measurements, i.e., the diurnal distribution of LLJ, the sensitivity of LLJ detection depending on the criteria and the properties of the detected LLJ events are presented in Section 3, followed by the comparison between model and measurement data. Conclusions of the discussed results are drawn in Section 4.

## 2. Data Sets

### 2.1. Simulated Data Set: ABL Model HIRVAC2D

The micro-scale model HIRVAC (HIgh Resolution Vegetation Atmosphere Coupler [27]) is used with two spatial dimensions to run the simulations. HIRVAC2D is suitable for carrying out extensive parameter studies and investigating the sensitivity of the unsteady flow field in the addressed height

range up to 500 m to changes in land use parameters. HIRVAC2D describes the exchange between the atmosphere, the vegetation, and the soil in vertical and horizontal direction. Different 1.5-order turbulence closure schemes can be used to solve the basic set of equations, e.g., the so-called *k-l* model, where *k* is the turbulent diffusion coefficient and *l* is the mixing length, which is diagnostically defined [27]. This approach to calculate the eddy diffusivity was used for the presented study according to Ziemann [28]. The prognostic model considers the inertial wind oscillation due to the impact of Coriolis force and resulting in the development of a nocturnal wind maximum above the inversion layer [9]. Additional terms in the basic equations of the model HIRVAC2D parameterize the exchange of momentum and heat between the vegetation elements and the environment. The set of basic equations is solved at about 100 vertical model layers, about the half of the layers inside higher vegetation like forests. In this way, daily courses of all meteorological variables can be numerically investigated in their variability throughout the year depending on the kind of land use and vegetation properties. A special feature of HIRVAC2D is to simulate wind fields transiently within the ABL and around realistic landscapes with forests and grasslands [29].

Hundreds of simulations were performed for a geostrophic wind speed $v_g = 4\,\mathrm{m\,s^{-1}}$ as well as for $v_g = 10\,\mathrm{m\,s^{-1}}$ and a cloudless summer day (Day Of the Year, DOY = 170) using several vegetation types and canopy parameter values, e.g., different profiles of plant area density (PAD) to describe the vertical distribution of vegetation. The needed PAD-data for vegetation parameterization can be derived from airborne or terrestrial laser scans which are applied meanwhile for wind energy applications [30]. An overview of the vegetation parameters and their values (e.g., typical PAD profiles for grassland, coniferous and deciduous forest) was provided together with all other model parameter values by Ziemann et al. [31]. For the further investigation, a height range between 50 m and 500 m was analysed, looking for the maximum wind speed. Vertical wind profiles were extracted from the simulated data base every 10 simulated minutes (without averaging). The following requirements for the wind profile were set as criteria for the occurrence of LLJs: Maximum wind speed $v_{max}$ superimposed a minimum wind speed $v_{min}$ (or profile value less than $v_{max}$), difference between $v_{max}$ and $v_{min}$ at least $0.5\,\mathrm{m\,s^{-1}}$ [32], ratio between $v_{max}$ and $v_{min}$ at least 1.05. The motivation for choosing this criterion was to discover also weakly developed LLJ events. Only the nocturnal simulations were analysed. The time interval of the study was between 8:00 p.m. and 8:00 a.m. local time (LT:CET) with at least 30 min duration of the event, i.e., LLJs that were too short during the night were not counted either. In addition, an LLJ event was not considered for analysis if, for example, it started at 11:00 p.m. and ended at 8:10 a.m. Condition for the event to be counted is that a time interval must be found before 8 a.m. Therefore, the high number of simulation runs was significantly reduced to about 500 for several values of vegetation parameters.

The HIRVAC2D runs for one set of initial conditions and vegetation parameters were performed over 48 simulated hours, frequently leading to more than one LLJ event during the considered nighttime periods. Only the longest LLJ event per simulation run was taken into account for the analysis. Almost always, the LLJ during the first simulated night was the longest one and was chosen for the analysis. In this way, the simulated data sets are comparable and a further drastic reduction of analysed simulation runs results: 293 (8) cases for $v_g$ of 4 (10) $\mathrm{m\,s^{-1}}$. Table 1 summarises exemplary the parameter values of vegetation height, plant area index (PAI; vertical integral of PAD) and vegetation cover (covered fraction of ground surface) for typical sites with grassland, deciduous and coniferous forest, respectively. All simulations with the chosen parameter sets led to LLJ events, especially for the grassland sites.

The HIRVAC2D data set is available on request from the corresponding author (A. Ziemann, TU Dresden).

**Table 1.** Exemplary parameter values of HIRVAC2D runs producing LLJ. Parameters: PAI is the plant area index (vertical integral of plant area density), vegetation cover is the vegetation-covered fraction of ground surface area. Combination of parameter values result in four types of grassland, two deciduous and one coniferous forests.

| Vegetation | Height in m | PAI | Vegetation Cover |
|---|---|---|---|
| Grassland | 0.1 | 3 and 1 | 0.9 and 0.5 |
| Deciduous forest | 15 or 28 | 5.2 or 6.0 | 0.9 |
| Coniferous forest | 37 | 9.4 | 0.9 |

### 2.2. Measured Data Set: Remote Sensing by Wind Lidar

Vertical profiles of the horizontal wind speed were obtained with the wind lidar WLS8-8 (Leosphere, France) of TU Braunschweig. It was operated at the airport Braunschweig at $10°33.27'$ E, $52°19.15'$ N, 91 m altitude, for an entire year, from 1 June 2013 to 31 May 2014. Wind data were recorded from 40 m to 500 m with a vertical resolution of 20 m. The temporal resolution of the data is 7 s. During the internal processing, data are averaged and stored as 10 min intervals. The whole data set with 10 min temporal resolution is publicly available [33] and statistics of LLJ occurrence have been described in [11]. There, the threshold criterion for identifying LLJ events is a wind speed difference of at least $2\,\mathrm{m\,s^{-1}}$ and 25% [7] between the wind speed maximum and the wind speed minimum above. Further, these criteria have to be met for at least 3 consecutive time steps, i.e., 30 min duration. This setup is called in the following the "$2\,\mathrm{m\,s^{-1}}$ criterion". Further, the same LLJ criterion as for the simulations (see Section 2.1) was applied to perform sensitivity tests, called here "$0.5\,\mathrm{m\,s^{-1}}$ criterion". For this model comparison, simulations were performed for meteorological conditions for one particular day in June (DOY = 170). Therefore, the focus of the measured data is on June 2013. For comparison, also the measurement data of July 2013 and August 2013 are analysed here.

## 3. Results

### 3.1. Characterisation of LLJ Properties Based on Wind Lidar Data Set

#### 3.1.1. Diurnal distribution of LLJ

LLJs can be expected between sunset and sunrise, with modifications by cloudy conditions. Figure 1 shows the diurnal frequency distribution of LLJ events according to the $2\,\mathrm{m\,s^{-1}}$ criterion for nighttime cases between 20:00 and 08:00 UTC (corresponding to 21:00 and 09:00 local time, CET) for the summer months June, July and August 2013.

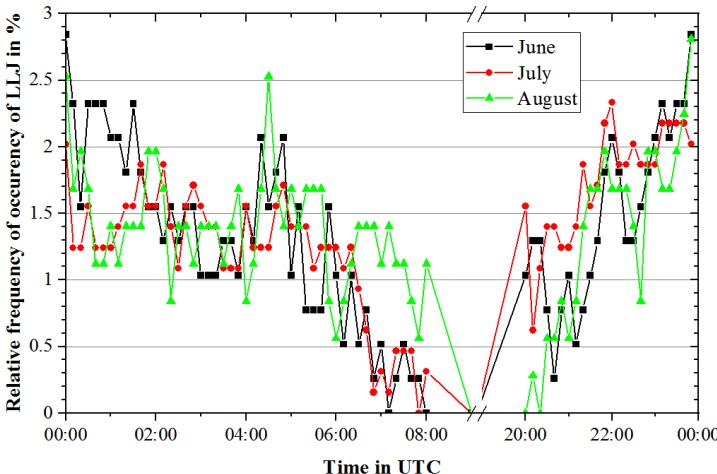

**Figure 1.** Diurnal distribution of LLJ events recorded at the study site Braunschweig, Germany, between 8 p.m. and 8 a.m. UTC for the months June 2013 (blue), July 2013 (red) and August 2013 (green).

LLJs are present throughout all night, but occur most frequently in the time range between 23:00 and 24:00 UTC. During the morning hours, the LLJ occurrence is also relatively high. This result is likely to indicate that long-term LLJ events with a duration over several hours have occurred. Please note, that all LLJ events within the considered height range are summed up. A long-lasting event, typically developing at smaller heights and rising up with the top of the inversion height contributes several times to the frequency distribution.

### 3.1.2. Sensitivity of LLJ Definition to Wind Speed Criteria

To avoid systematic errors by comparing measured and simulated LLJ events based on different criteria, the $0.5\,\mathrm{m\,s^{-1}}$ criterion was also applied to the measurement data. For the $2\,\mathrm{m\,s^{-1}}$ criterion, the overall number of 10 min intervals of LLJ occurrence in June 2013 was 387, resulting in a total time of around 64 h influenced by LLJ events. For the $0.5\,\mathrm{m\,s^{-1}}$ criterion, the number of 10 min intervals of LLJ was 996, which is a total time of 166 h. Besides the large differences in absolute numbers of LLJ occurrence, the relative distribution of LLJ occurrence depending on altitude (Figure 2) and maximum wind speed (Figure 3) is different: For the stronger criterion requiring a wind speed difference of at least $2\,\mathrm{m\,s^{-1}}$, the most frequent altitude of the LLJ maximum is in the range of 180 m to 300 m. As an expected result for the lower criterion, the most frequent occurrence of LLJ events is at lower altitudes, in the range of 40 m to 80 m. Please note, that the distribution has several peaks.

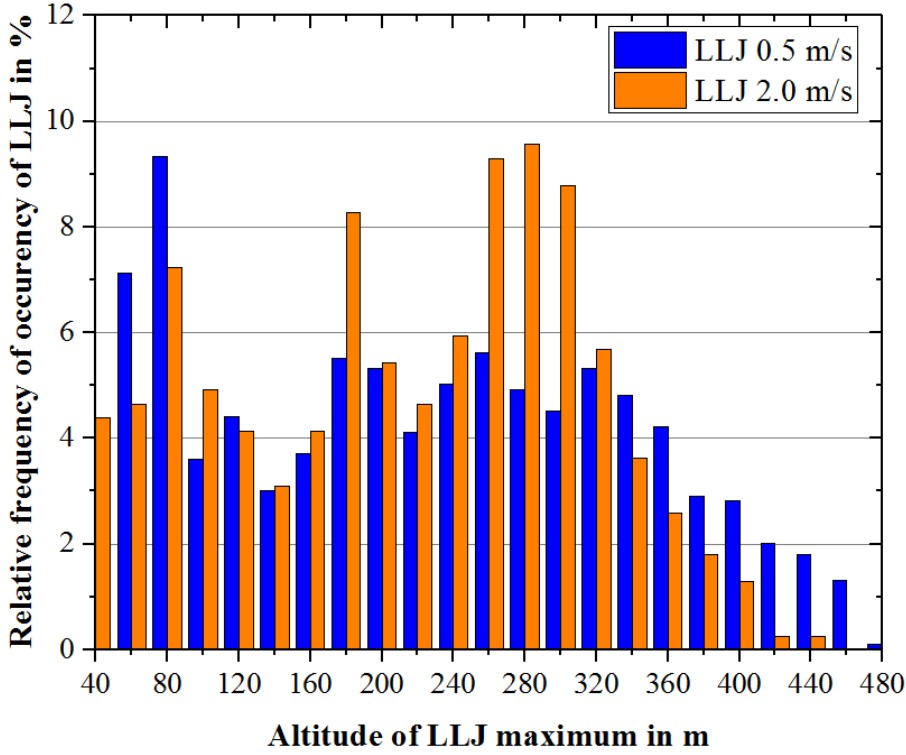

**Figure 2.** Comparison of LLJ events according to the criteria of $2\,\mathrm{m\,s^{-1}}$ (orange) and $0.5\,\mathrm{m\,s^{-1}}$ (blue) for the months June, July and August 2013.

For the stronger criterion requiring a wind speed difference of at least $2\,\mathrm{m\,s^{-1}}$, the most frequent maximum wind speed of the LLJ is in the range of $6\,\mathrm{m\,s^{-1}}$ to $10\,\mathrm{m\,s^{-1}}$ (Figure 3). For the criterion requiring only a wind speed difference of $0.5\,\mathrm{m\,s^{-1}}$, the frequency of occurrence has a broader distribution. Altogether, stronger LLJ events appear to have the maximum at higher altitudes. This is investigated in the following section.

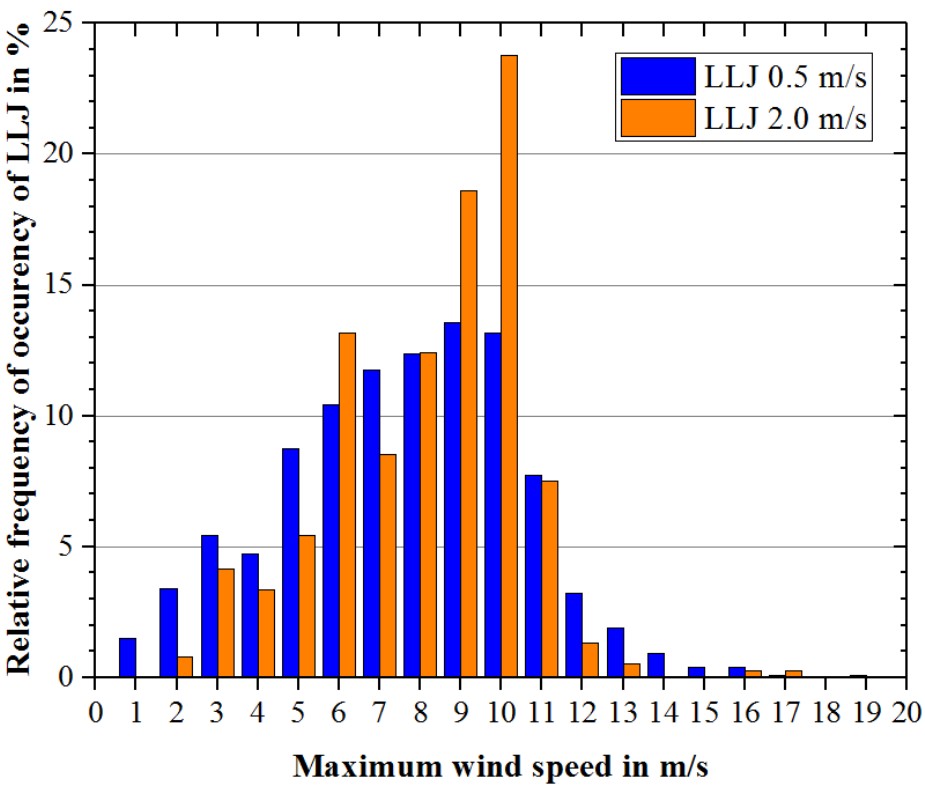

**Figure 3.** Comparison of LLJ events according to the criteria of $2\,\mathrm{m\,s^{-1}}$ (orange) and $0.5\,\mathrm{m\,s^{-1}}$ (blue) for the months June, July and August 2013.

### 3.1.3. LLJ Altitude and Wind Speed

An increase of the LLJ maximum wind speed with the LLJ height was observed for all measurements (Figure 4). Generally, for the course of time of consecutive wind profiles, the wind speed and maximum tend to increase with time. For all three investigated summer months in 2013, an almost linear increase of the maximum wind speed and LLJ altitude was observed especially in the wind speed range up to about $10\,\mathrm{m\,s^{-1}}$. In this range, a rather continuous increase of altitude was observed. The clearest relationship between increasing maximum wind speed and increasing height of the LLJ can be found in August. It is assumed that the increasing nighttime in August leads to more pronounced development of temperature inversions and to more low-level jet events due to the mechanism described by Blackadar [9].

For higher wind speed values, the LLJ could also occur at lower altitudes. It is not yet clear which special causes are responsible for these LLJ events. The break in the near of $10\,\mathrm{m\,s^{-1}}$ is perhaps triggered by the smaller probability of occurrence for stability-driven LLJ events at higher levels of geostrophic wind speed and forcing. This result coincides with simulation results: there are only a few LLJ events and at a rising altitude for $v_g = 10\,\mathrm{m\,s^{-1}}$ in comparison to $v_g = 4\,\mathrm{m\,s^{-1}}$ (see next section). There was no detected LLJ in the model runs at $v_g = 20\,\mathrm{m\,s^{-1}}$. Additionally, there are a few measured LLJ events at very high altitudes. Lifted temperature inversion layers, i.e., without connection to ground surface, may exist and lead to those LLJ events.

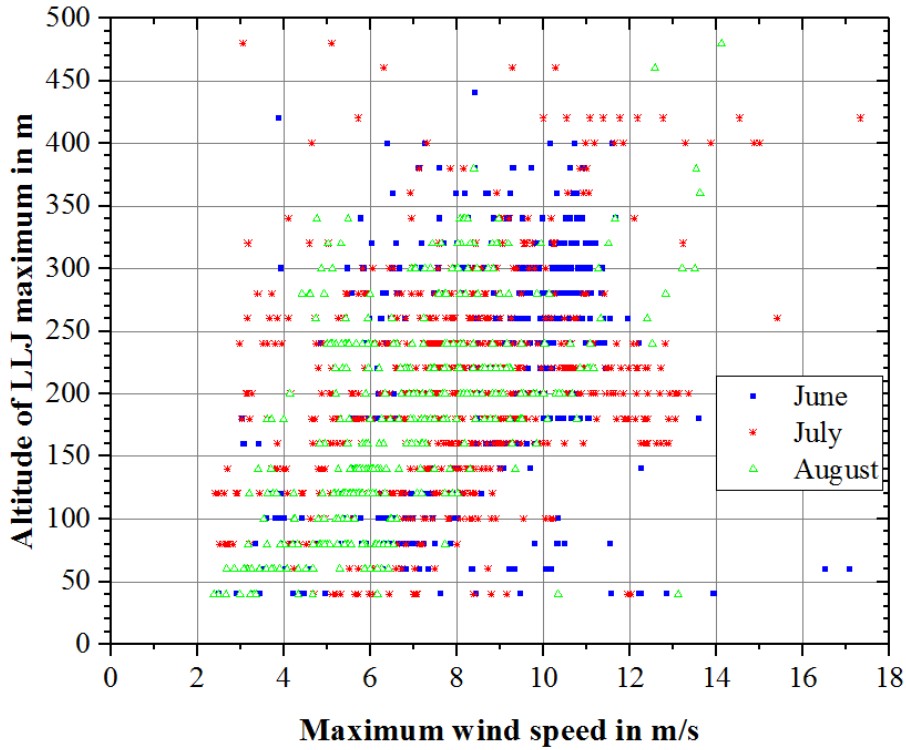

**Figure 4.** Relationship of maximum wind speed and altitude of maximum wind speed for LLJ events identified by the $2\,\mathrm{m\,s^{-1}}$ criteria during the months June 2013 (blue), July 2013 (red) and August 2013 (green).

*3.2. Comparison of Measured and Simulated Data Set*

3.2.1. Development of Single LLJ Events

Figure 5 shows the temporal development of a particular typical exemplary LLJ event on 18/19 June 2013. The wind speed is normalised with the maximum wind speed observed at core height (320 m altitude) at 6 UTC. The LLJ development begins with a maximum at low altitude (160 m–180 m). During the night, and probably associated with an increasing ground-based temperature inversion, the maximum rises to higher altitudes and gets more intense with an almost linear increase of the wind speed with altitude. Finally, after sunrise, the temperature inversion is probably lifted up from the surface, and the core height is lifted as well. The intensity of the event (maximum wind speed) begins to decrease.

One example for a typical simulated LLJ is shown in Figure 6. The temporal development of LLJ begins in the early morning hours shortly after initialization of the model run. As in the measurements, the core height of the LLJ increases with time due to the rising height of ground-based temperature inversion. At 7 LT (local time, CET) the maximum wind speed is reached at an altitude of about 180 m for the smaller $v_g$-value of $4\,\mathrm{m\,s^{-1}}$ and 380 m for $v_g = 10\,\mathrm{m\,s^{-1}}$. The connection between increasing altitude and increasing wind speed at core height is more pronounced for the higher geostrophic wind speed. This result is in agreement with the shown example of a measured LLJ (Figure 5 ). As the ground-based temperature inversion begins to lift and to weaken at 8 LT, the LLJ is also rising and the maximum wind speed is decreasing for both geostrophic forcings (Figure 6). In contrast to measurements, the typical shape of the vertical profile of the simulations is different: The vertical wind shear is greater in the measurements than in the simulations, especially above the LLJ core. This behaviour is probably connected to a different development and intensity of temperature inversion as well as different turbulence characteristics above the inversion. To check these relationships and to provide additional data for model comparison, it is desirable to carry out future wind measurements together with measurements of the temperature and turbulence profile.

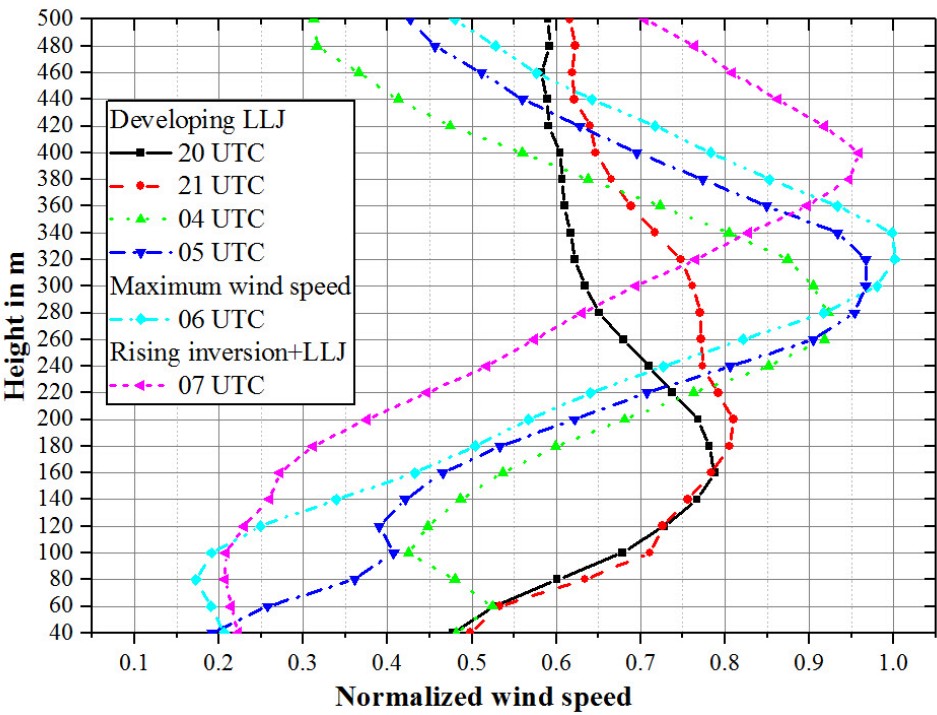

**Figure 5.** Vertical profiles of wind speed normalized by maximum value at the LLJ core ($11.2\,\mathrm{m\,s^{-1}}$), measured by wind lidar during the night 18–19 June 2013.

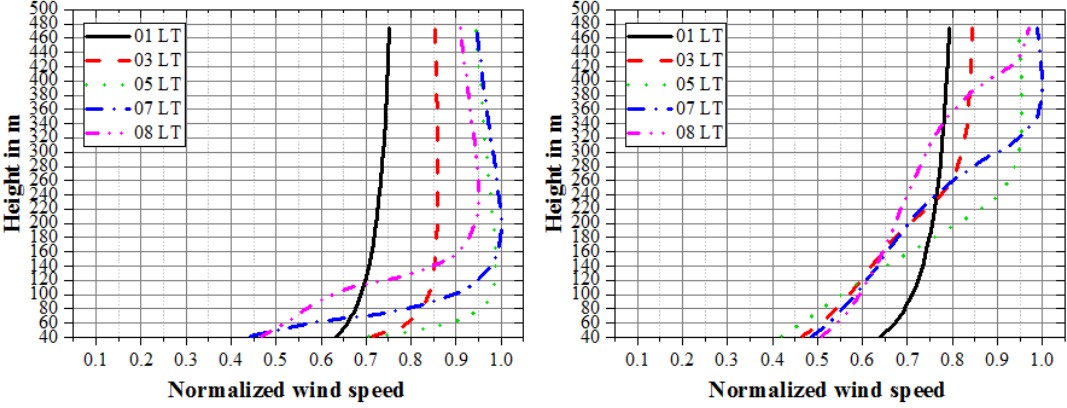

**Figure 6.** Vertical profiles of wind speed, left: $v_g = 4\,\mathrm{m\,s^{-1}}$, right: $v_g = 10\,\mathrm{m\,s^{-1}}$. Wind speed was normalized by maximum value at the LLJ core (left: $5.2\,\mathrm{m\,s^{-1}}$, right: $12.0\,\mathrm{m\,s^{-1}}$) simulated by HIRVAC2D during a night (LT: local time, CET) in June (DOY = 170) over grassland, parameter values see Table 1 (PAI = 1, vegetation coverage = 0.9).

### 3.2.2. Properties of LLJ in Measured and Simulated Data

Figures 7 and 8 show a comparison of measured and simulated LLJ properties for the months of June and July 2013. The x-axis refers to the maximum values of the wind speed, while the y-axis corresponds to the maximum height of LLJs detected by the $0.5\,\mathrm{m\,s^{-1}}$ criterion. For a better differentiation of the simulated results, different symbols were implemented for the HIRVAC2D simulations of grassland for a geostrophic wind speed of $4\,\mathrm{m\,s^{-1}}$ and $10\,\mathrm{m\,s^{-1}}$, the simulations of coniferous forest and deciduous forest, both for $v_g = 4\,\mathrm{m\,s^{-1}}$. In contrast to grassland, no LLJ events with a maximum wind speed up to 500 m altitude were detected over forested landscapes for $v_g = 10\,\mathrm{m\,s^{-1}}$.

The different lidar data for June and July were used to demonstrate that the simulated data coincide with the spanned value range of measurements. There are more LLJ events with smaller wind speed values in July 2013 so that the simulations for $v_g = 4\,\text{m}\,\text{s}^{-1}$ closer agree with measurements.

Please note, that the measurements in 2013 were not performed aiming at an atmospheric model evaluation and vice versa. The initial conditions for the HIRVAC2D simulation runs, e.g., initial vertical profiles of wind vector and temperature as well as turbulence parameters, are not adapted to the real conditions due to a not available complete data set of various measurements. Against this background, the presented results of the comparison between measurement and model results are to be regarded as a first assessment and are intended to provide information which additional measurements are necessary for a further model evaluation.

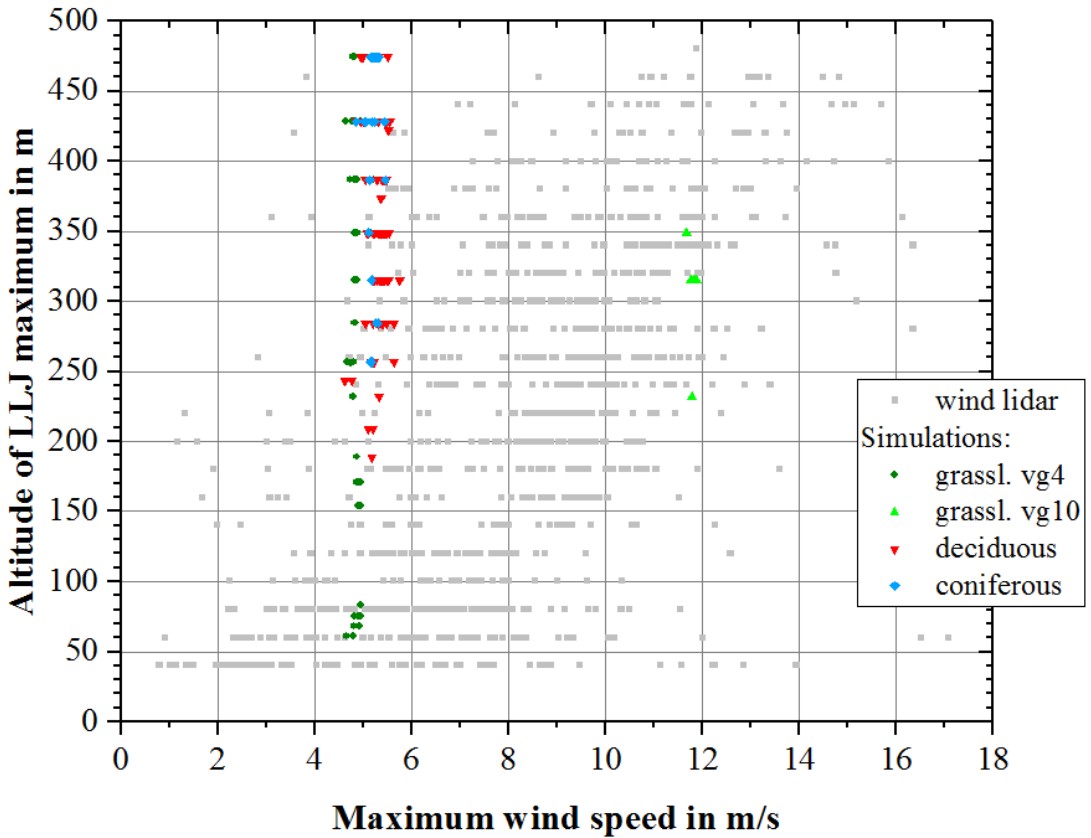

**Figure 7.** Comparison of maximum height and wind speed of measured (light grey) and simulated (grassland for $v_g = 4$ and $10\,\text{m}\,\text{s}^{-1}$: dark and light green, deciduous forest: red, coniferous forest: blue) LLJ events detected with the $0.5\,\text{m}\,\text{s}^{-1}$ criterion for June 2013.

Furthermore, Figures 7 and 8 show that the number and the height of the LLJ depends on the geostrophic wind speed $v_g$. As the $v_g$ values increase, the LLJ altitude also increases. An increasing $v_g$ leads to a decreasing number of LLJ events, especially for high and dense forest stands. The altitude of LLJ core is rising up for grassland sites resulting in fewer LLJ events that occur in considered height range. This increase of the LLJ altitude together with rising maximum wind speed was also observed in the lidar measurements. The fact that the height of the LLJ depends on the wind speed has also been observed by other authors, e.g., Rausch et al. [34].

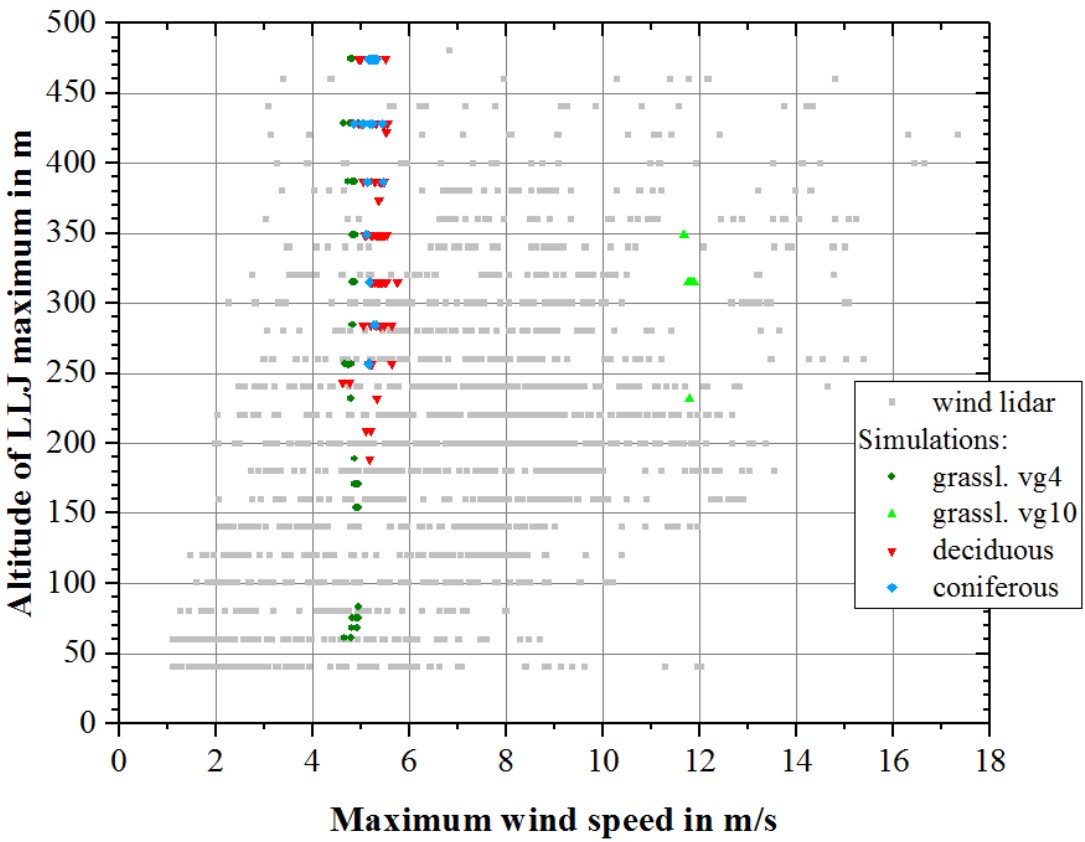

**Figure 8.** Comparison of maximum height and wind speed of measured and simulated LLJ events detected with the $0.5\,\mathrm{m\,s}^{-1}$ criterion (same as Figure 7) for July 2013.

## 4. Discussion and Conclusions

In the study, it was shown that important characteristics of the frequent phenomenon of low-level jets, which is of particular interest for wind energy, can be measured by a wind lidar and represented in HIRVAC2D simulations. According to the theory of Blackadar [9], LLJs form when the boundary layer is decoupled from surface friction by a temperature inversion. Temperature inversions typically build up after sunset and during the night. This behaviour was be detected in measurement and simulation data. The further investigation of the data focused on the height and maximum wind speed of LLJ.

Analyses of lidar measurements obtained during summer 2013 at the flat grassland site of the airport Braunschweig showed a typical increase of maximum wind speed with altitude, especially in the wind speed range below about $10\,\mathrm{m\,s}^{-1}$. This is the case both for individual LLJ events and for the whole 3-months statistics.

Numerical simulations with HIRVAC2D are able to capture this general behaviour of increasing wind speed maximum and altitude. The lidar data were used to demonstrate that the simulated data coincide with the spanned value range of measurements. In July 2013, more LLJ events with smaller wind speed values occurred resulting in a closer agreement with the simulations for the relatively small value of $v_g = 4\,\mathrm{m\,s}^{-1}$. Additionally, the comparison of model data with measurements showed that boundary conditions have to be adapted in simulation runs to provide realistic LLJ properties. It was found that the kind of land use (grassland or deciduous/coniferous forest) and vegetation parameters (vegetation height, coverage of ground surface and compactness of canopy) are important for the LLJ prognosis. It can be observed that the maximum wind speed of LLJs is on average higher for forests than for grasslands. According to [31], the higher the altitude of the vegetation (grasses, trees), the higher is also the height of the LLJ. This result is related to the fact that the altitude of the LLJ (and therewith the maximum wind speed) depends on the roughness height and in this way on

the land use model and the vegetation parameters. Otherwise, the lower the altitude and compactness of the vegetation (i.e., value of PAI and vegetation coverage), the lower the altitude of the LLJ. It was also observed that for a grassland site, the nocturnal LLJ is noticeably more frequent in the considered height range. Although the airport Braunschweig is best classified as a grassland site, it has to be noted that it is partly surrounded by a mixture of deciduous and coniferous forest, in particular to the East and South. In future model evaluation it is planned to include this realistic horizontal heterogeneity and to consider this influence on the LLJ characteristics. Principally, HIRVAC2D is prepared to simulate heterogeneous land surfaces [29].

In contrast to the agreement in LLJ altitude and maximum wind speed, the wind shear cannot be reproduced realistically. One possible reason is the used 1.5-order closure approach for the turbulence closure. The so-called *k-l* model was used for the HIRVAC2D simulations, where *k* is the turbulent diffusion coefficient and the *l* is the mixing length which is diagnostically defined [27]. For the presented results, a certain mixing length approach was applied (Ziemann [28], Equations (6)–(8)). Other mixing length parameterizations are possible ([28], Equation (9)). Furthermore, approaches derived from turbulence measurements in forest stands (e.g., Queck et al. [29]) can be applied. In addition, parameterizations for the mixing length can be derived from measurements at other locations (e.g., Sogachev et al. [35], Sogachev [36]), which may be more suitable for comparison with the present wind lidar measurements. In addition to various mixing length approaches, a closure approach using a prognostic equation for the dissipation of the turbulent kinetic energy is built into HIRVAC2D. This approach could also lead to a better match between the simulation and measurement results.

The uncertainties of the model results depend on several factors with decreasing impact:

- the possibility to directly simulate significant processes for the object of investigation,
- the parameterizations in the model,
- the data for the model initialization,
- the data for land use and vegetation parameters,
- the choice of boundary conditions,
- the numerical uncertainty regarding the model-inherent time step and the choice of numerical method.

It is not possible to make a general statement about the model uncertainty for all possible model applications. The model uncertainty can only be derived from a coordinated comparison with natural observations and measurements. The results of the present study provide initial qualitative indications for similarities and differences of the wind profile at altitudes of more than 50 m. The data base is not sufficient for a full model validation at the moment. Thereby, the uncertainty and the representativeness of the wind lidar measurements have to be taken into account. The uncertainty of the wind lidar measurements is in the range of less than $0.5 \, \mathrm{m \, s^{-1}}$ according to the manufacturer. The measurements of wind speed with wind lidar is a well-established technique which has been compared and validated against other systems like cup anemometers extensively (e.g., [37]). The agreement of wind speed measurements is very good, not taking into account turbulence. However, the question is how representative the measurement for larger areas. The LLJ has been observed to occur above large areas of several 10 km (e.g., [38]). This spatial variability on the scale of kilometers might be substantial.

For future development of the model HIRVAC2D, more LLJ events in the measured and simulated data sets should be compared. Additionally, the verification of atmospheric stability by temperature profiles is of high relevance for a realistic simulation of the LLJ. Therefore, the wind lidar measurements should be complemented with other continuous measurement techniques, like a radio acoustic sounding system (RASS) or a microwave profiler. There are measurement sites that offer such a selection of remote sensing devices, e.g., at the Meteorological Observatory Lindenberg—Richard-Assmann Observatory (MOL-RAO) of the German Weather Service. As these measurement systems are not flexible enough, an alternative could be regular temperature measurements with unmanned aerial systems, e.g., [39–41].

Both the presented wind measurements and the microscale model simulations can help to improve wind potential investigations at individual locations including features such as increased wind shear (e.g., [4,20,26,42]). In addition to these studies for long climatological periods of 10 to 20 years, model applications for larger areas and shorter periods, e.g., short-term forecasts for the next 24 h, are possible. Microscale models like HIRVAC2D may close the gap between the present mesoscale output of weather models and the necessary model resolution to describe phenomena like the low-level jet. In practice and in particular in Africa, there is a need for methods to infer the wind speed at typical hub heights from existing measurements on 10-m-high weather masts [43]. A model such as HIRVAC2D can also be used for this purpose of data extrapolation.

**Author Contributions:** A.Z. assessed the HIRVAC2D simulation runs. A.G.A. did the comparison of LLJ based on simulations and measurements in the framework of his bachelor thesis, supervised by A.Z. and A.L. A.L. acquired the wind lidar data set. All authors contributed to the manuscript and commented on the text. All authors have read and agreed to the published version of the manuscript.

**Funding:** The HIRVAC2D simulations were funded by the project QuWind100 (German Federal Ministry for Economic Affairs and Energy, BMWi, on the basis of a decision by the German Bundestag, funding code 0325940A). The lidar measurements and the further research received no external funding. The publication was funded by the publication funds of TU Dresden.

**Acknowledgments:** The authors would like to thank Manuela Starke for performing the HIRVAC2D simulations. The authors would like to acknowledge the two anonymous reviewers for proof reading the manuscript.

**Conflicts of Interest:** The authors declare no conflict of interest.

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
