# Peer review of "Comparison of Wind Lidar Data and Numerical Simulations of the Low-Level Jet at a Grassland Site"

_energies, doi:10.3390/en13236264_

Round 1

Reviewer 1 Report

The paper entitled “Comparison of wind lidar data and numerical simulations of the low-level jet at a grassland site” focuses on the numerical investigation of the low-level jet at a grassland site. Even though the paper deals with an interesting topic, some questions should be addressed before the publication in journal Energies.

  1. The authors should explain how to improve the numerical model in order to adequately describe the wind velocity profile above the LLJ location?
  2. Can the authors point out the novelty of the performed research compared to already published papers?
  3. How do the authors plan to incorporate spatial distribution of plants within the numerical model?
  4. Verification and validation of the obtained results is missing within the manuscript. What is the numerical uncertainty of the obtained results?
  5. What is the experimental uncertainty of the measured wind data?
  6. The conclusions are not supported by the obtained results.

Reviewer 2 Report

The paper presents an interesting research which fits to the scope of the journal. However, there are some issues which should be corrected before publishing. Below I present comments which should be considered while improving the paper.

  1. Please expand the introduction not only by the description of historical development of wind energy in German condition but also by future energy transition policies and goals (e.g. Sustainable Development Goals, European Green Deal, etc.).
  2. Figure 1: I suggest to change the type of chart from bar chart to line chart. In the current version it is difficult to distinguish patterns for analyzed months due to the density of bars.
  3. For better visualization the input of this study to the state of art I suggest to consider splitting results from discussion, so that comparative discussion highlighting differences or similarities of this research to other studies are presented in clearer way.
  4. I suggest to consider highlighting in conclusions the role of more accurate wind potential assessment, especially as recent and up-to-date studies refer quite commonly to more generalized datasets describing wind energy potential (e.g. An Assessment of Wind Energy Potential for the Three Topographic Regions of Eritrea, Energies 2020; Where Renewable Energy Sources Funds are Invested? Spatial Analysis of Energy Production Potential and Public Support, Energies 2020; Platform Optimization and Cost Analysis in a Floating Offshore Wind Farm, J. Mar. Sci. Eng. 2020). The improvement of data presented in your study would enable to improve other studies in different aspects of renewable energy domain.
  5. Submitted manuscript has relatively limited literature review (32 references), however, improvement of elements mentioned above could be helpful to improve it.

I encourage the Authors to correct the paper, as in my opinion it presents an interesting study and might constitute a valuable paper after improvements mentioned above.

Round 2

Reviewer 1 Report

The authors have adequately addressed most of the comments and extended the discussion and conclusion part significantly. Novelty of the proposed research is now pointed out.

The authors have explained in detail how to include the heterogeneity of vegetation and wind speed profile above the location of LLJ in the numerical model.

However, even though the possible sources of numerical uncertainty are now emphasized, the verification of the numerical results is still missing. Despite that, in my opinion the manuscript is acceptable for publication in the present form.

Reviewer 2 Report

The paper has been corrected according to my previous comments and in my opinion it can be published in the current form.